# A Systematic Review of Atopic Dermatitis: The Intriguing Journey Starting from Physiopathology to Treatment, from Laboratory Bench to Bedside

**DOI:** 10.3390/biomedicines10112700

**Published:** 2022-10-25

**Authors:** Giulia Radi, Anna Campanti, Federico Diotallevi, Emanuela Martina, Andrea Marani, Annamaria Offidani

**Affiliations:** Dermatological Clinic, Department of Clinical and Molecular Sciences, Polytechnic Marche University, 60126 Ancona, Italy

**Keywords:** atopic dermatitis, pathophysiology, small molecules, biologics, novel treatments

## Abstract

Atopic dermatitis (AD) is a common chronic inflammatory and immune-mediated skin disease with a complex pathophysiology and still represents a therapeutic challenge, owing to limited responses to available treatments. However, recent advances in the understanding of AD pathophysiology have led to the discovery of several new potential therapeutic targets, and research in the field of new molecules with therapeutic perspectives is boiling, with more than 70 new promising drugs in development. The aim of this systematic review is to provide the state of the art on the current knowledge concerning the pathophysiology of the disease and on novel agents currently being investigated for AD, and to highlight which type of evolution is going to take place in therapeutic approaches of atopic dermatitis in the coming years.

## 1. Introduction

Atopic dermatitis (AD) is an inflammatory skin disease characterized by a complex interplay of genetic, environmental and immunological factors. Multifactorial etiopathogenesis is also responsible for the extreme clinical variety of AD. For several years, AD has been considered primarily a childhood disease; it has a prevalence in children ranging from 2.7% to 20.1% among different countries and from 2.1% to 4.9% in adults [1], with different phenotypes both for the clinical presentation and mode of onset, from persistent forms from childhood to late onset forms in adulthood or in the elderly [2]. In recent years, the study of the pathogenetic pathway of atopic dermatitis has made it possible to better understand the pathophysiology of the disease and to identify new key molecules with a practical impact in the therapeutic field. The aim of this review is to provide an overview of the new molecular targets starting from the careful observation of the pathophysiological mechanisms of AD up to the future development of new therapies that are increasingly targeted and tailored as much as possible to the different phenotypes of AD patients.

## 2. Materials and Methods

This systematic review was conducted following the approach developed by Arksey and O’Malley [3,4], which consists of five key steps: the identification of the research question; the identification of adequate studies; the selection of studies; the monitoring of data; and the collection, summary and reporting of results. The Preferred Reporting Items for Systematic Reviews and Meta-Analysis (PRISMA) extension for scoping review criteria was used to conduct the review [4].

### 2.1. Identification of the Research Question

A brainstorming approach involving the entire research team was used to identify the research questions. The research group included four dermatologists with expertise in atopic dermatitis.

At the initial meeting, the research group identified the research question and determined the research strategy. The research question was: “How can increasing knowledge in the pathophysiology of atopic dermatitis drive treatment developments”?

### 2.2. Study Selection Process

We performed a worldwide systematic review of studies reporting on the pathophysiology and treatment of AD, using three electronic medical databases: PubMed, EMBASE and the Web of Science, considering articles dated 1 May 2012 through 1 May 2022 (the last ten years).

The search terms were selected to identify studies describing the relation between the pathophysiology and treatment of AD. 

The keywords used were “pathophysiology AND atopic dermatitis”, “pathophysiology AND atopic dermatitis AND biologics”, “pathophysiology AND atopic dermatitis AND small molecules” and “pathophysiology AND atopic dermatitis AND novel treatments”.

All selected databases were searched from their beginning, without any time interval. Furthermore, we searched by hand the reference lists of other relevant articles on the pathophysiology and treatment of AD. 

In the first step, 238 records were screened from the selected databases. There were 200 records after the removal of duplicates. Among the selected records, none were marked as ineligible. The relevant studies were then selected. This process occurred in three stages. In the first stage, four researchers (G.R., A.C., F.D. and E.M.) independently selected articles based on title. Any disagreements were resolved by consulting a senior researcher (AO). In the second stage, two members of the research team (G.R. and A.C.), evaluated each abstract independently. The research group resolved all discrepancies through a unanimous consensus. A total of 83 articles were excluded, and 117 were evaluated for a full-text analysis. Of these, 21 manuscripts were not retrieved.

The third step consisted of a critical evaluation of the full text of the 96 selected papers. In the end, 55 studies were selected for the final qualitative synthesis. To be included in our review, studies had to be focused on the pathophysiology of AD, on a novel therapeutic approach to AD and on the efficacy and safety of novel molecular target treatments for AD, both in pediatric and adult patients. All included studies had to be in English with the abstract available. No restrictions on study design were considered, and randomized controlled trials, case-control studies, cross-sectional studies and case series were included. Articles were excluded from our review for only three reasons: reason 1 = studies on animal models (12 reports were excluded for this reason), reason 2 = articles generically related to allergic diseases (17 reports were excluded for this reason) and reason 3 = reports in languages other than English (12 reports were excluded for this reason). 

### 2.3. Data Extraction

Before data extraction, A.C. designed a data extraction form to speed up the whole process. To answer the research question, the following information was extracted from the included articles: author(s) name and publication date; study design; study population; sample size; measured outcomes; study results; and study recommendations.

## 3. Results

The flowchart of the PRISMA study is shown in Figure 1. Our search identified 200 records after eliminating duplicates. After a review of the titles and abstracts, 104 citations were removed, and 96 were evaluated for full-text eligibility. After the full text review, 55 case-control studies, case series studies, randomized controlled trials and meta-analysis studies were found to be eligible and included in this study. 

The data found show that increased knowledge in the physiopathology of AD has been fundamentally changing its treatment paradigm and will do so for years to come. Moreover, increased clinician experience in the treatment of AD has led to understanding how different pathophysiological pathways may in turn contribute to determining AD development.

## 4. From the Bench: AD Pathophysiology

Dysfunction of the epidermal barrier and immune dysregulation are known to play a reciprocal role in the pathogenesis of AD. Due to AD complexity and heterogeneity both on a clinical and molecular level, the primum movens of the AD pathogenetic process is far from being fully understood. 

Genetic factors play a crucial role in the etiopathogenesis of the disease, as confirmed by the evidence of a family predisposition [5]. Genes previously associated with AD encode factors in the innate and adaptive immune systems as well as proteins that regulate the terminal differentiation of keratinocytes [6]. To better understand the pathophysiology of the disease, we therefore tried to give a transversal overview of all the different components that contribute to defining atopic dermatitis and which translate into targets for the development of new therapeutic tools.

### 4.1. Epidermal Barrier Dysfunction

The skin barrier exerts a physical and chemical protective action; it prevents allergen penetration, it maintains skin hydration and it displays antimicrobial activity, while having an ongoing interaction with the immune system. Differentiated terminal keratinocytes; structural proteins such as loricrin (LOR), involucrin (INV) and filaggrin (FLG); and lipids are components of the cornified envelope (CE), which is primarily responsible for the mechanical integrity of the barrier. An acidic skin pH [5,6,7] is necessary to maintain the integrity of the CE, lipid metabolism, epidermal differentiation and antimicrobial functions [7]. Several mechanisms contribute to the skin barrier damage in AD: alterations of FLG or LOR synthesis, changes in the metabolism of lipids and membrane ceramides, the loss of the integrity of tight junctions and the dysregulation of the differentiation process of keratinocytes. They can be genetically determined or induced by interaction with the immune system [8,9]. 

### 4.2. Skin Microbiome

The human skin microbiome is dominated by several phyla of bacteria (Firmicutes, Proteobacteria, Bacteroidetes and Actinobacteria). The cutaneous microbiome of subjects affected by AD is characterized by reduced diversity of bacterial strains both in non-lesional skin, and it is even more evident in AD lesional skin compared with the skin of healthy subjects [10]. AD skin is characterized by the increased colonization of Staphylococcus Aureus (SA), which can also be attributable to barrier damage (for example, resulting from an altered metabolism of membrane lipids). Other coagulase-negative staphylococcus strains, Staphylococcus epidermidis (SE) and Staphylococcus hominis (SH) are also increased in AD skin but are less than SA growth. This causes them to lose their potential to balance the microbial action performed by SA, which becomes pathogenic. 

Under physiological conditions, SE and SH can produce antimicrobial peptides (AMPs) [10], and SA acts as a trigger for the activation of the local inflammatory response, involving both the innate and adaptive immune systems. SA penetration across the epidermis directly correlates with increased interleukin (IL)-4, IL-13, IL-22 and TSLP and with decreased expression of AMPs [11]. Moreover, it contributes to barrier damage. Similarly, the gut microbiome study of AD patients also reveals significant differences compared to the microbiome of healthy controls [12].

### 4.3. Immune Dysregulation

Immune dysregulation involves both innate and adaptive immune responses. It can also be genetically determined for barrier damage by genetic mutations or epigenetic modifications in the cells involved. It certainly plays a key role in the pathogenesis of the disease, and it is as important as it is complex. 

Keratinocytes are protagonists both in the pathophysiology of barrier damage and in the activation of the innate immune response. They act as a sentinel, as they are the first to receive the insults, whether they are mechanical or inflammatory, and they answer by activating and producing AMPs and proinflammatory cytokines. The AMPs, especially cathelicidin (LL-37) and human-b-defensins 2 and 3, act on tight junctions to restore the skin barrier dysfunction and exert an autocrine function on keratinocytes, which start to release pro-inflammatory cytokines also called “alarmins”: IL-25, IL-33 and thymic stromal lymphopoietin (TSLP). The alarmins activate the innate lymphoid cell 2 (ILC2) and other dermal lymphoid cells (dendritic cells (DCs), Langerhans) which in turn produce IL-5 and IL-13 and promote the differentiation of the type two adaptive immune response by triggering a self-reinforcing inflammatory loop [13].

## 5. To the Bedside: New Molecular Targets 

### 5.1. Cutaneous Microbiota 

Reduced skin microbiota diversity has been demonstrated in patients with AD. There is an imbalance towards the colonization of *Staphylococcus aureus* to the detriment of other commensals such as *Staphylococcus epidermidis*, *Staphylococcus hominis* and *Roseomonas mucosa*, which, in physiological conditions, exert a protective role through the production of antimicrobial peptides [13]. SA is often responsible for infectious complications and, in addition, acts as a trigger for cutaneous inflammatory responses both by stimulating Th2 lymphocytes to produce proinflammatory cytokines such as IL-4 and IL-13, and by simultaneously activating the Th17 pathway and the synthesis of IL-22 [14]. Rebalancing the skin microbiota can lead to a reduction in the SA load and can downregulate the inflammatory response triggered by SA. Microbiome transplantation and bacterial replacement are being considered for several new topical treatments as the targets of interest for the microbiome rebalancing to antagonize SA colonization. In detail, the commensal strains of coagulase-negative SE and SH are employed for transplantation, and commensal Gram-negative *coccobacillus Roseomonas* mucosa act by anti-inflammation via Toll-like receptor 5 (TLR5) and tumor necrosis factor receptor (TNFR) activation as a bacterial replacement [15]. A spray product containing *Nitrosomonas eutropha* is under Phase II. *Nitrosomonas eutropha*, an ammonia-oxidizing bacterium able to produce nitric oxide, is a mediator with anti-inflammatory and metabolic properties. The application of a topical ointment of 2% niclosamide (ATx201) has also been shown to be effective in reducing SA colonization [16]. Finally, another approach consists of oral integration with preparations of bacterial strains with therapeutic potential capable of altering the intestinal microbiome, reprogramming the metabolism, reducing immune activation and thus protecting against allergic inflammation. The EDP1815, STMC-103H and KBL697 are in Phase I-II for AD. The clinical trials targeting microbiome molecules in AD are shown in Table 1. 

### 5.2. The Alarmins: TSLP, IL-25 and IL-33

TSLP, IL-25 and IL-33, known as “alarmins”, are molecules of the innate immune system produced by keratinocytes [17]. The synthesis of alarmins generally occurs following an insult suffered by the keratinocytes, which are activated and trigger the local inflammatory response. TSLP, IL-25 and IL-33 can recruit the Langerhans cells resident in the skin that activate the Th lymphocytes, polarizing the Th2 adaptive immune response [18]. Therefore, considering alarmins as early activators of the epithelial Th2 immune response, several trials have been conducted to evaluate the efficacy of antibodies directed against these molecules for the treatment of AD. Among these, the antibody directed against TSLP and an IL-33 inhibitor have been shown to be effective in the treatment of asthma.

#### 5.2.1. TSLP

TSLP is a molecule involved both in physiological and in pathological processes, mainly under inflammatory conditions. It is physiologically produced by epithelial cells of the skin, lungs and gastrointestinal tract and plays a crucial role in the homeostasis of Treg lymphocyte differentiation mediated by DCs and plasmacytoid DCs (pDCs). In inflammatory conditions, other immune cells such as DCs, basophils, and mast cells can also produce TSLP, promoting Th2 cytokine responses. TSLP binds to a heterodimeric receptor of DCs consisting of the IL-7 receptor α-chain (IL-7Rα) and the TSLPR chain activating the STAT5 intracellular signaling pathway. TSLP stimulates DCs to upregulate co-stimulatory molecules such as OX40L, CD80 and CD86 and can drive the activity of IL-4, IL-5 and IL-13-producing CD4^+^ T cells.

TSLPR and IL-7Rα transcripts are also expressed in human dorsal root ganglion cells, suggesting that keratinocyte-derived TSLP can be a therapeutic target against pruritus in AD [19].

A Phase II randomized controlled study evaluated the efficacy of the anti-TSLP monoclonal antibody in association with TCS on moderate-to-severe atopic dermatitis [20]. The study ended showing a greater but not statistically significant improvement in eczema compared to the placebo group.

#### 5.2.2. IL-33

IL-33 is an alarmin cytokine produced by several different cell types, including epithelia, endothelia, fibroblasts and hematopoietic cells such as mast cells, neutrophils, monocytes, macrophages and dendritic cells, following cell damage resulting from infections or inflammation. IL-33 binds to a heterodimeric receptor consisting of the ST2 and IL-1 receptor accessory protein (IL-1RAcP), triggering MyD88, interleukin-1 receptor-associated kinase 1 (IRAK1) and IRAK4 with the downstream activation of nuclear factor kB (NF-kB) or mitogen-activated protein kinase (MAPK) pathways. IL-33 polarizes the innate and adaptive type two immune response by promoting the activation of ILC2, eosinophils, Th2 and the IL-8-induced migration of neutrophils to the sites of infection. It also stimulates the synthesis of proinflammatory cytokines such as IL-4, IL-5 and IL-13. The proof-of-concept clinical trial showed a clinical improvement after a single intra- venous 300 mg dose of etokimab, with a humanized IgG1/kappa anti-IL-33 monoclonal antibody in 12 adult patients with moderate-to-severe AD [21]. A Phase IIb study on an anti-IL-33 monoclonal antibody for the treatment of AD (REGN3500) has recently been discontinued due to a loss of efficacy. These findings suggest the role of IL-33 in the pathogenesis of AD or other type two diseases such as asthma (GSK3772847) and the therapeutic potential of IL-33 inhibition for their treatment. Therefore, further studies are needed to confirm the effectiveness in the treatment of AD. The clinical trials targeting TSLP and IL-33 in AD are shown in Table 2.

### 5.3. Interleukin 1α (IL-1α)

IL-1α is a molecule produced by keratinocytes during stressful situations such as physical or chemical insults or infections. It has a pro-inflammatory action and, in detail, it is involved in the antigen presentation and in the induction of the inflammatory cascade [22]. It has been considered a potential therapeutic target in both oncological and inflammatory diseases. The efficacy results of Phase II studies of monoclonal antibodies against IL-1α in atopic dermatitis have been poor, and they appear promising in hidradenitis suppurativa [23]. The clinical trials targeting Interleukin 1α in AD are shown in Table 3.

### 5.4. The Epidermal Xenobiotic Receptor (AhR) (2)

Keratinocytes together with other skin cells such as fibroblasts and dendritic cells express several chemical sensors, such as the aryl hydrocarbon receptor (AHR) AHR, also called the dioxin receptor. It binds to environmental polyaromatic hydrocarbons and dioxins with high affinity and activates the AHR-nuclear translocator (ARNT) system with a dual function. On one hand, it responds to the insult by strengthening the skin barrier functions and accelerating terminal epidermal differentiation through the upregulation of the expression of filaggrin; on the other hand, it induces oxidative stress by generating abundant reactive oxygen species (ROS). Moreover, AHR also activates antioxidative transcription factor NRF2 and upregulates the expression of Phase II antioxidative enzymes [24].

As a crucial chemo sensor, AHR activity modulates immune functions in the skin and has been identified as a potential therapeutic target in inflammatory skin diseases such as AD. Several trials are underway to demonstrate the efficacy of a topical cream product containing an AhR agonist for the treatment of AD in adults and children [25,26]. The same molecule is also being studied as a topical treatment for psoriasis [27]. The clinical trials targeting AhR in AD are shown in Table 4.

### 5.5. Antigen Presentation OX40-OX40L

Antigen presentation is a crucial step in the human adaptive immune response. By this procedure, T cells recognize molecules towards which they direct a specific immunological response through the activation and clonal expansion of lymphocytes. It occurs in conditions of both type two and type one inflammation and in antitumoral processes. Therefore, since antigen presentation can be a potential therapeutic target in inflammatory diseases, capable of modifying the natural history of the disease, it is nevertheless necessary to monitor the long-term safety profile, especially in those patients with an increased risk of cancer. OX40 (also known as CD134 or TNFRSF4) and its ligand OX40L have been identified among the molecules involved in antigen presentation in AD [28]. OX40 belongs to the TNF receptor family (TNFRS4) and is a secondary costimulatory molecule on the surface of activated T cells, and OX40L is expressed on DCs. In chronic inflammatory diseases such as AD, the activation of the OX40L-OX40 axis triggers and holds the response of T lymphocytes over a long term, playing a key role both in the onset phases and in the chronicization of the disease. Several trials are ongoing to evaluate the efficacy and safety of monoclonal antibodies directed against OX40 (ISB 830 and KHK4083) and OX40L (KY1005). The clinical trials targeting OX40 in AD are shown in Table 5.

### 5.6. Phosphodiesterase 4 Inhibitors (PDE4)

Phosphodiesterase 4 (PDE4) is the enzyme responsible for the degradation of cyclic AMP (cyclic adenosine 3′,5′-monophosphate (camP)), a second key messenger which plays a role in the synthesis of several cytokines involved in the inflammatory cascade in pulmonary, neurological, rheumatoid, gastrointestinal and dermatological disorders. By targeting PDE4, by its inhibition, intracellular AMPc levels increase with a consequent reduction in the synthesis of pro-inflammatory cytokines, and, at the same time, there is a promotion of the synthesis of anti-inflammatory cytokines [29]. PDE4 has been recognized since the 1980s as a therapeutic target in the dermatological field for the treatment of inflammatory skin diseases. Apremilast, a small molecule PDE4 inhibitor, has recently been approved as an oral therapy for the treatment of moderate-to-severe psoriasis, and it has not been shown to be effective in the systemic treatment of AD. However, the topical application of PDE4 inhibitors has shown efficacy in the treatment of mild-to-moderate eczema without the side effects of systemic administration, affecting the gastrointestinal system. Topical drugs containing PDE4 inhibitors can be placed into the therapeutic algorithm as an alternative to the use of topical corticosteroids (TCS) or calcineurin inhibitors such as tacrolimus or pimecrolimus. Crisaborole has already received approval with the indication for the treatment of mild-to-moderate AD. In addition, a new trial for characterizing the role of crisaborolo in skin microbiome changes is ongoing (NCT04800185). Other studies are underway to develop new topical PDE4 inhibitors (Roflumilast, lotamilast and difamilast) and new oral PDE4 inhibitors with a high affinity to subtype PD4 (Orismilast). The clinical trials targeting PDE4 in AD are shown in Table 6.

### 5.7. Interleukin 22

IL-22 is an interleukin produced by T cells, primarily owing to the stimulation induced by staphylococcal exotoxin. Keratinocytes are the main targets of IL-22, which trigger the cellular proliferation and the downregulation of filaggrin expression. IL-22 exerts a pro-inflammatory action in both psoriasis and AD [30]. The difference between the two activation pathways appears to be due to the T cell source. In psoriasis, the cytokine is predominantly released by CD4 cells [31], whereas CD8 cells appear to be the primary producers of IL-22 in AD. Several trials are ongoing to evaluate the efficacy of monoclonal antibodies directed against IL-22 and its receptor (IL-22R) for the treatment of moderate-to-severe AD. An interesting aspect of targeting IL-22 is that the serum levels of IL-22 seem to be higher in patients with severe AD, and their response to treatment is better than in subjects with moderate AD [32]. This suggests that IL-22 could be a potential biomarker of disease severity, or that it could in some way allow us to type the patient with AD and provide him with more targeted therapy. The clinical trials targeting IL-22 in AD are shown in Table 7.

### 5.8. Interleukin 17C

IL-17C is an epidermal cytokine primarily produced by keratinocytes. The synthesis of IL-17C is induced both in conditions of skin inflammation and of a bacterial infectious stimulus via the TLR2 and TLR5 pathway. By binding the IL-17A and IL-17E receptors, which are expressed in multiple cell types, including keratinocytes and T lymphocytes, IL-17C performs an autocrine action on keratinocytes by inducing the synthesis of pro-inflammatory cytokines and a paracrine action on lymphocytes T, promoting autoimmune activity with the production of IL-17A/F and IL-22 [33]. IL17C appears to play a role in the pathogenesis of both psoriasis and AD, particularly in patient phenotypes (including Asian AD, intrinsic AD and pediatric AD) that exhibit higher tissue IL-17 expression. Considering IL-17C as a potential therapeutic target in AD can be interesting given the great clinical heterogeneity of AD, which reflects a variable interaction on the molecular level of the Th2 pathway with other cytokines more expressed in Th1-type pathways such as IL-17 and IL-22. It can be a resource for patients who do not achieve complete or near-complete disease resolution with current therapies targeting Th2 cytokines. Trials that tried to demonstrate the efficacy of an anti-IL17c monoclonal antibody were stopped for futility; however, this therapeutic target could be reconsidered following the stratification of patients and after having outlined, where possible, a more careful molecular profile of some AD phenotypes, starting, for example, with the non-responding patients to target therapies that are currently available. The clinical trials targeting IL-17c in AD are shown in Table 8.

### 5.9. Histamine 4 Receptor Antagonists (H4R)

The histamine 4 receptor (H4R) is a member of the histamine receptors family composed of four different types: H1R, H2R, H3R and H4R, and they are all are G-protein-coupled receptors. Each one is expressed on different types of cells and tissues with its own biological effect. In the central nervous system (CNS) and on the skin, we can find H1R, which mediates itches; H2R regulates the contractility of the smooth muscle of the gastrointestinal tract; and H3R regulates the synthesis of histamines in the CNS. H4R is expressed in several cell types of the immune system and epithelial cells. It is assumed to play an important pro-inflammatory role in various diseases, including bronchial asthma, atopic dermatitis and pruritus [34]. It has been considered a therapeutic target for AD, and oral H4R antagonist molecules are currently in Phase II [35]. The clinical trials targeting H4R in AD are shown in Table 9.

### 5.10. Interleukin 31

IL-31 is a proinflammatory cytokine involved in the pathogenesis of AD by affecting barrier function and in the transmission of pruritus by triggering dorsal root ganglia and nerve innervation. IL-31 levels correlate with AD severity. Together with IL-4 and IL-13, it is produced mainly by Th2 cells and exerts its pruritogenic action both by acting directly on the damage of the skin barrier and by activating the neuroimmune circuit following binding with the IL-31R receptor present on C nerve fibers. IL-31 and IL-31R binding causes the heterodimerization of IL-31Rα and the oncostatin M receptor-β (OSMRβ) to release an itching sensation [36]. Given its involvement in the complex mediation of independent histamine pruritus, it soon became an object of study as a potential therapeutic target for AD, but it also became an object of study for inflammatory skin diseases characterized by intense itching such as prurigo nodularis [37]. Several studies are underway on monoclonal antibodies that antagonize both heterodimer components of the receptor. Nemolizumab is an anti-IL-31Rα that is near commercialization for moderate-to-severe AD, Vixarelimab is an anti-OSMRβ that targets IL-31 and oncostatin-M (OSM) is currently in a Phase II study for prurigo nodularis. Studies on Nemolizumab are also ongoing in the pediatric population with AD and pruritus associated with chronic kidney disease (NCT05075408) and systemic sclerosis (NCT05214794). The clinical trials targeting IL-31 in AD are shown in Table 10.

### 5.11. Substance P-NK-1R

Substance P is another molecule involved in the transmission of itching. It appears to have an influence on mast cell degranulation, but the exact mechanism through which it exerts its pruritogenic action is more complex and mediated by the activation of its receptors expressed on nerve fibers: neurokinin 1 receptor (NK1R) and the Mas-related G-protein-coupled receptor (MRGPRS) [38]. It seems that inflammatory skin diseases such as AD and prurigo nodularis, characterized by chronic and intense itching, show a high expression of substance P and NK1R, so these molecules can be considered as therapeutic targets [39]. Several trials were conducted on two small molecule inhibitors of NK-1R, Tradipidant and Serlopidant, to evaluate their efficacy on the resolution of pruritus in AD and PN (Table 11).

### 5.12. P2X Purinoreceptors 3 (P2RX3)

P2X purinergic receptors (purinoceptors) are a family of cation-permeable, ligand-gated ion channels that open in response to the binding of extracellular adenosine 5′-triphosphate (ATP). The P2X purinoceptor family includes seven members (P2RX1–P2RX7). The P2X3 receptor ion channel is expressed by sensory or autonomic neurons and plays an important role in peripheral irritation, pain sensation and coughing, but it may potentially play a role in itching as well [40]. The P2X3 receptor has been identified as target in AD pruritus, and a selective P2X3 antagonist (BLU-5937, is a new oral small molecule that is under study for the evaluation of its efficacy on the itch improvement of chronic pruritus in adult subjects with atopic dermatitis. The clinical trials targeting the P2X3 receptor in AD are shown in Table 12.

### 5.13. Anti-IgE Therapy

The role of IgE in the pathogenesis of AD is somewhat controversial. Patients with atopic dermatitis show variable blood levels of IgE. In particular, subjects with atopic comorbidities such as polysensitivity to aeroallergens or foods have higher IgE levels, and contact with these allergens can also be responsible for the exacerbation of dermatitis. It is not clear whether IgE-mediated allergic reactions to environmental allergens can become irrelevant after a long duration of the disease, particularly in adults, if, on the other hand, late seroconversion can occur over time in subjects affected by AD with low values of IgE at the onset of the disease. In the extrinsic forms with childhood onset, the presence of high IgE values can be considered an early or late aspect in the pathogenesis of AD, also considering the higher incidence of food and respiratory allergies in children with atopic dermatitis compared to non-atopic ones. Beyond the serum levels of IgE, the expression of high-affinity IgE receptors (FcεRI) on epidermal Langerhans cells is a characteristic biomarker for AD [41,42,43]. Targeting IgE serum levels and high-affinity IgE FcεRI can be useful to identify a subtype of patients with AD with a practical implication on the therapeutic approach as well. However, studies conducted so far in AD with Omalizumab, an anti-IgE monoclonal antibody, indicated for chronic spontaneous urticaria (CSU), allergic asthma and chronic rhinosinusitis with nasal polyps (CRSwNP), have been found to be inconclusive [44,45]. The best results in terms of efficacy on eczema have been recorded in patients with total IgE serum levels not exceeding 700 IU/mL. In those subjects with very high IgE values, associating plasmapheresis with omalizumab was tried with better results than single biological therapy. A new high affinity monoclonal antibody for IgE, ligelizumab, is under study for CSU, and it will be interesting to see if it could have therapeutic implications also in selected patients with AD. Another approach inherent in IgE activity is to interfere with its synthesis by targeting B cells. Under physiological conditions, B cells express a membrane form of IgE (mIgE), which binds other human IgE + cells including IgE-committed lymphoblasts and memory B cells. Treatment with anti-CεmX mAbs allows for the activation of B cell receptor (BCR) signaling, causing anergy and antibody-dependent cell cytotoxicity (ADCC) of the mIgE + cells. This results in a drop in IgE-producing plasma cells, hence resulting in reduced IgE synthesis, which, over time, leads to a desensitized state in allergic patients [46]. The clinical trials targenting IgE in AD are shown in Table 13.

### 5.14. IL-4 and IL-13

AD is an inflammatory skin disease in which an imbalance of the Th1/Th2 immune response occurs with a bias towards type two immunity. This occurs especially in the acute phases of the disease, and with the chronicization of AD, the lesional skin expresses a significant Th1 component in addition to Th2 responses. The Th2 cytokines, such as IL-4, IL-5 and IL-13, have specific effects on the epidermis, including the suppression of keratinocyte differentiation and AMP production, which contribute to the AD skin phenotype [47]. IL-4 and IL-13 are the crucial Th2 cytokines involved in AD pathogenesis; they contribute to disease pathology by driving distinct and overlapping effects. IL-13 is a key cytokine known to be an important stimulator of peripheral inflammation by the activation of effector cells, such as eosinophils, and by tissue remodeling (leading to fibrosis and smooth muscle changes) [48]. IL-4, driving Th2 cell differentiation, seems to be more focused on the early steps of AD pathogenesis. In addition, IL-4 plays a main role in IgE class switching. Although IL-4 signaling is involved in the pathogenesis of AD, its selective inhibition has not led to useful therapeutic findings [49]. By targeting both the IL-4 and IL-13 pathways, AD therapy has undergone a breakthrough. In March 2017, the FDA approved dupilumab, the first fully human IgG4 mAb for moderate-to-severe AD, which opened the door to use mAb and other biologics in the treatment of AD. Dupilumab inhibits IL-4 and IL-13 signaling by binding to the alpha subunit of the IL-4 receptor (IL-4Rα), shared with IL-13R [50]. The therapeutic advantage of blocking this receptor subunit is to contemporarily neutralize the signaling of both IL-4 and IL-13 that show overlapping but not of redundant functions in potentiating type two inflammation. Several monoclonal antibodies are currently being developed. CBP-201 and AK120 are antibodies directed against IL-4Rα that are currently in Phase Ib and II studies. ASLAN004, currently in Phase Ib, is a fully humanized antibody directed against IL-13Rα1, thereby blocking the binding of IL-4 and IL-13 on type two receptors (IL-4Rα/IL-13Rα1). IL-13 antibodies such as tralokinmab and lebrikizumab are close to commercialization. Tralokinumab is a fully humanized antibody targeting IL-13 that blocks its binding to both IL-13Rα1 and IL-13α2 receptor chains. It obtained FDA approval for AD in December 2021. Lebrikizumab is another fully humanized anti-IL-13 antibody that does not block the binding of the cytokine to the receptor but instead impairs the heterodimerization of IL-4Rα and IL-13Rα1, thereby inhibiting signal transduction. It is now under Phase III studies for AD. The clinical trials targeting IL-4 e IL-13 in AD are shown in Table 14.

### 5.15. IL-5

Blood eosinophil levels have been shown to increase in AD in some patients. Since IL-5 is a Th2 cytokine that is important for eosinophil recruitment, it has been considered to be a target for the development of a monoclonal antibody, inhibiting IL-5 activity. However, results from a Phase II trial have revealed that the human IL-5 Ab, mepolizumab, does not show efficacy in patients with moderate-to-severe AD [51]. A case report of a successfully treated atopic dermatitis in a patient with severe, uncontrolled asthma under Benralizumab (a monoclonal antibody targeting IL-5Rα) has been reported [52]. The clinical trials targeting IL-5 in AD are shown in Table 15

### 5.16. JAK Inhibitors: New Beacons of Hope in the Treatment of Atopic Dermatitis

Janus kinases (JAKs) are intracytoplasmic protein tyrosine kinases (TYKs) that bind to the cytoplasmic region of transmembrane receptors of cytokine of types one and two and mediate cellular responses to numerous cytokines and growth factors. These mediators are involved in immune defense and immune-mediated diseases as atopic dermatitis [53,54]. The interaction of the cytokine receptor with a ligand, such as a cytokine or growth factor, induces activation of receptor-associated JAK proteins, leading to tyrosine phosphorylation of the receptor, which then leads to activation of transcriptional signal transducers and activators (STATs) [55]. STAT proteins translocate into the nucleus and modulate the transcription of effector genes. Cellular responses to JAK/STAT signaling can therefore include proliferation, differentiation, migration, apoptosis and cell survival and, in the case of the abnormal activation of a particular pathway, they can induce the development of autoinflammatory diseases such as atopic dermatitis [56]. There are four isoforms of JAK: JAK1, JAK2, JAK3 and TYK2, which mediate signaling in pairs specifically associated with particular cytokine and growth factor receptors, most often involved in the pathogenesis of atopic dermatitis [57,58].

JAK1 is important for the signaling of receptors activated by IL 6, IL-10, IL-11, IL-19, IL-20, IL-22 and interferon (IFN) alpha, IFN-beta and IFN-gamma. JAK1 can be paired with any of the other three members of the JAK family, depending on the associated receptor.

JAK2 is important for signaling the hormone cytokines erythropoietin, thrombopoietin, growth hormone, granulocyte-macrophage colony-stimulating factor (GM-CSF), IL-3 and IL-5. JAK2 can couple with JAK1, TYK2 or another JAK2 molecule, depending on the associated receptor.

JAK3 is expressed in hematopoietic cells and is activated when the corresponding cytokine receptor for IL-2, IL-4, IL-7, IL-9, IL-15 and IL-21 binds to the respective ligand. These cytokines are important for lymphocyte activation, function and proliferation. JAK3 signals only in combination with JAK1.

TYK2 couples with JAK1 or JAK2 or another TYK2 molecule to facilitate IL-12, IL-23 and type one IFN signaling. JAKs are not involved in the process of lymphocyte activation [57,58].

Several pharmacological JAK inhibitors (JAKi or Jakinib) are commercially available for clinical use as oral and topical agents for immune-mediated and inflammatory diseases. Specifically, oral formulations include abrocitinib, upadacitinib, baricitinib and gusacitinib and are indicated for patients with moderate-to-severe AD. The emerging topical formulations under development include ruxolitinib and deglocitinib. Topical agents can be used in patients with localized AD and also in addition to systemic therapy in patients with more severe, not-well-controlled diseases [59].

Abrocitinib is an oral selective JAK1 inhibitor approved in Europe, the United States and Japan for the treatment of moderate-to-severe atopic dermatitis in adults whose disease is not controlled with other systemic therapies, including biologics, or when the use of such therapies is not indicated. Two Phase III trials mainly enabled the drug’s approval, with one of them comparing Abrocitinib with dupilumab [60,61]. The results are shown in Table 16.

Upadacitinib is an oral selective JAK1 inhibitor approved in the United States, Europe and other countries for the treatment of adults with AD and in children older than 12 years when other systemic drugs, including biologics, have failed, or when the use of such therapies is contraindicated.

Several multicenter, randomized trials have assessed the efficacy of upadacitinib in adults and adolescents with moderate-to-severe atopic dermatitis as a monotherapy or with topical corticosteroids, compared with a placebo [62,63]. Upadacitinib, compared with Dupilumab in a head-to-head study, demonstrated greater efficacy with a larger percentage of patients achieving the primary endpoint of EASI 75 at 16 weeks (71 vs. 61%) and a significant reduction in pruritus (55% vs. 36%) [64].

Baricitinib is an oral small molecule JAK (1-2) inhibitor approved for the treatment of moderate-to-severe atopic dermatitis in Europe and the USA that block multiple cytokine signaling, including IL 4, IL-5 and IL-13, involved in the pathogenesis and progression of the disease. The BREEZE-AD program includes several Phase III trials, which have shown the efficacy and safety results of baricitinib, compared to the placebo, in monotherapy [65] or in association with topical corticosteroids. A head-to-head study with dupilumab is not available.

Topical ruxolitinib is a JAK inhibitor that was approved in September 2021 by the U.S. FDA for the short-term treatment of mild-to-moderate atopic dermatitis in immunocompetent patients older than 12 years with a disease not controlled by topical therapies. Two eight-week randomized Phase III trials with an identical design (TRuE-AD1 and TRuE-AD2) evaluated the efficacy of topical ruxolitinib for the treatment of mild-to-moderate AD [66].

Other topical JAK inhibitors that are under study are tofacitinib and deglocitinib. The efficacy of topical tofacitinib for the treatment of atopic dermatitis was evaluated in a randomized Phase IIa study [67]. Delgocitinib, an experimental topical pan-Janus kinase (JAK) inhibitor, inhibits all members of the JAK family (JAK1, JAK2, JAK3 and tyrosine kinase 2). The topical formulation is approved in Japan [68] for the treatment of atopic dermatitis and is also used in clinical development for alopecia areata, discoid lupus erythematosus, psoriasis inversa and atopic dermatitis in the USA and Europe [69].

Gusacitinib (ASN002) is an oral inhibitor of SYK and JAK signaling, including Tyk2. The inhibition of SYK-JAK by gusacitinib modulates Th2, Th22, Th1 and Th17 cytokines and regulates IL17 keratinocyte-mediated signaling and proliferation/differentiation. Thus, gusacitinib can simultaneously target immune and epithelial cells involved in chronic hand eczema and other dermatologic and inflammatory conditions such as atopic dermatitis [59].

Gusacitinib was evaluated in patients with moderate-to-severe atopic dermatitis in a 4-week randomized double-blind placebo-controlled study, resulting in superiority to the placebo in the percentage of patients achieving EASI50 and in the change from the baseline in itching [69]. The results are shown in Table 16.

## 6. Discussion

Atopic dermatitis (AD) is a chronic inflammatory and immune-mediated disorder with an incidence ranging from 20% in children and 10% in adults depending on region and ethnicity. Due to its severe pruritus and chronicity course, AD has been shown to have a great psychosocial burden on patients’ lives [70].

It has been largely reported that AD impairs quality of life, raises stigmatization levels and frequently leads patients to develop secondary psychiatric comorbidities, such as depression and anxiety. Moreover, several data from the literature attempt to demonstrate the systemic nature of AD as a result of the increased evidence of both the associated allergic and non-allergic comorbidities [70].

Starting from these considerations, today, dealing with AD is more than “treating chronic eczema”. The therapeutic strategies available and those that will appear soon have the ambition to modify not only the clinical course but also the natural history of the disease, preventing the cumulative burden that the disease has on the lives of affected patients [71,72]. Understanding AD pathogenesis in depth may represent a starting point that is useful for understanding how the therapeutic panorama of AD is being enriched, through a “bench to bedside” journey [72].

According to the “outside-in” theory loss of function filaggrin gene polymorphisms, together with other mutations, involving other barrier-related proteins, such as involucrin and loricrin, should be primarily involved in the pathogenesis of AD. However, 40% of the carriers of filaggrin null alleles never experience eczema, and the contribution of the other barrier protein dysfunction in AD pathogenesis is far from being demonstrated [73].

On the other hand, it has been reported that filaggrin mutations are associated with AD severity and disease persistence in adulthood, suggesting that filaggrin mutations are responsible for a particular endotype of AD [74]. In addition, genetic and acquired defects in tight junctions have been reported to play a crucial role in barrier dysfunction in AD patients.

Current areas of study include the correlation between tight junction disruption and antigen presentation, the interaction between claudin and filaggrin expression and the effect of claudin expression on keratinocyte proliferation and differentiation [75].

According to the “inside-out” hypothesis, skin inflammation precedes barrier impairment and can lead to an impaired skin barrier. It has previously been reported that inflammatory states weaken the barrier by downregulating filaggrin production in the skin [76].

The effects of monoclonal antibodies or JAK-inhibitors on the Th2 inflammatory response and filaggrin downregulation and normalization in AD patients is a field that is currently under study. However, although the pathogenesis of AD is complex and only partially understood, beyond the “primum movens” of disease, it is becoming ever clearer that what should be suppressed is the interaction model between “inside” and “outside”, to obtain good control of the disease over the long term [77].

It has been reported that decreased expression in claudin 1 is associated with an increase in serum biomarkers of the Th2 driven response, suggesting cross-talk between the epithelial barrier and immunological inflammation in AD [78].

The role of microbiota dysbiosis is emerging in AD pathogenesis. *SA* triggers the host immune system-related inflammation in the acute phase of disease, and the persistence of *SA* in skin biofilm has also been already demonstrated in the chronic phase of AD. SE, although is considered an innocent bystander, may sometimes contribute to the development of inflammation in AD pathogenesis. *Cutibacterium acnes* enhances *SA* cytolytic activity, promoting the secretion of several pro-inflammatory cytokines. Moreover, *Malassezia* can elicit a specific IgE response as part of AD pathogenesis. The role of *Candida albicans* in promoting inflammation or immunological reactions of AD still remains debated [79].

Starting with these considerations, a complementary approach to AD treatment, which involves the use of probiotics, prebiotics or skin microbiota transplantation, is emerging; however, the efficacy of these novel therapeutic interventions is still being confirmed.

Very promising new treatments that are emerging in the AD therapeutic scene are topical JAK-inhibitors with Ruxolitinib cream, which has recently received FDA approval for the treatment of mild-to-moderate AD in adults. This is a crucial step in topical AD therapy, since the introduction of new topical effective and safe principles in the etherapeutic management of AD marks a crucial step towards true innovation of the topical management of AD, which, until now, in Europe, is mainly based on the use of steroids and topical calcineurin inhibitors [80].

Starting with the encouraging therapeutic results obtained from trials on topical Ruxolitinib, several other topical agents such as Tapinarof, Difamilast and Roflumilast are also in advanced stages of development. The long term efficacy and safety of these formulations, together with comparisons with current topical treatments, represent one of the most intriguing fields of research for near future [81,82,83].

Alongside these innovative research fields, approaches based more classically on the use of molecularly targeted drugs capable of modifying the inflammatory pathway of AD, acting on several key molecules, such as IL-1α, IL5, IL17, IL25, TLSP, IL31 and IL33, are being researched. Again, the progression in knowledge of the processes underlying the inflammatory cascade has led to the identification of specific and effective molecular targets, whose blockade may improve many features of disease, including pruritus, inflammation, skin barrier disruption and skin colonization [84,85].

In the preliminary phase, studies that aim to evaluate the effect of targeting pruritus through the inhibition of P2X purinoreceptors 3, neurokinin 1 receptor or substance P are also interesting [86,87,88].

This type of molecular approach appears more symptomatic than those based on cytokine blocking, but this therapeutic strategy could be promising in modifying the vicious circle related to pruritus in chronic AD.

There are more contradictory results based on blocking IgE, and this reflects a lack of knowledge in the role of IgE-mediated allergic reactions to environmental allergens in chronic AD, particularly in adults [46].

Results of completed studies on the latest effective systemic and topical drugs used to treat atopic dermatitis are available respectively in Table 17 and Table 18.

In conclusion, therapeutic approaches to AD have been globally changing over the last five years as a result of a progressive increase in knowledge on the pathogenetic mechanisms underlying the disease. It is reasonable to foresee a therapeutic future characterized by a personalized approach to AD patients, since molecules capable of producing therapeutic effects based on the patient’s phenotype will be available. In this scenario, the real challenge will be, for the years to come, to improve access to novel therapies for an increased number of patients. Results from complete studies of the newest topical drugs used for the treatment of atopic dermatitis are shown in Table 18.

## Figures and Tables

**Figure 1 biomedicines-10-02700-f001:**
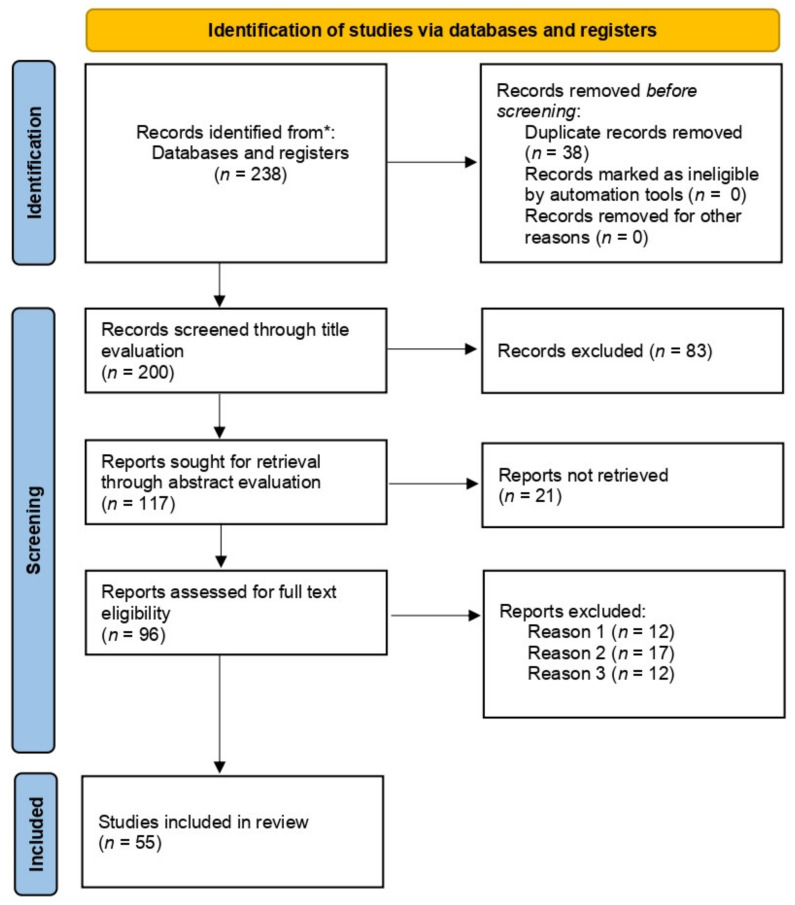
Preferred Reporting Items for Systematic Reviews and Meta-Analysis (PRISMA). * From: [4].

**Table 1 biomedicines-10-02700-t001:** Clinical trials targeting skin microbiome molecules in AD.

Target Molecule	Clin Trial Gov	Type of Study	Status
**Microbiome**			
**Topical bacterial strains**			
Targeted Microbiome Transplant Lotion (TMT)	NCT03151148	Phase I/II for AD	Completed
Autologous Microbial Transplant	NCT03158012	Phase I for AD	Completed
Autologous Microbial Transplant	NCT01959113	Phase I for AD	Completed
Three strains of Roseomonas mucosa FB-401	NCT04504279	Phase II for AD	Completed
Three strains of Roseomonas mucosa FB-401	NCT04936113	Phase II for AD	Terminated (Failure of the Phase II study (protocol FB401-01) to meet its endpoint))
Lyophilized strain of Staphylococcus hominis A9 (ShA9) (ADRN-UCSD-001)	NCT05177328	Phase I for AD	Recruiting
*Nitrosomonas eutropha spray* (B244)	NCT04490109	Phase II for AD	Completed
**Topical small molecule**			
Topical niclosamide 2% (ATx201)	NCT04339985	Phase II for AD	Completed
Topical niclosamide 2% (ATx201)	NCT03304470	Phase II for AD	Completed
Synthetic antimicrobial cationic peptide Omiganan (CLS001)	NCT02456480	Phase II for AD	Completed
Synthetic antimicrobial cationic peptide Omiganan (CLS001)	NCT03091426	Phase II for AD	Completed
**Oral bacterial strains**			
EDP1815	NCT05121480	Phase II for AD	Recruiting
STMC-103H	NCT05003804	Phase I-II for AD	Recruiting
KBL697	NCT04056130	Phase I for AD	Completed

**Table 2 biomedicines-10-02700-t002:** Clinical trials targeting TSLP and IL-33 in AD.

Target Molecule	Clin Trial Gov	Type of Study	Status
**TSLP**			
Anti-TSLP antibody Tezepelumab (AMG-157/MEDI9929)	NCT02525094	Phase IIa for AD	Completed
**IL-33**			
Anti-IL-33 antibody Etokimab (ANB020)	NCT03533751	Phase IIa for AD	Recruitment status unknown
Anti-IL-33 antibody Itepekimab (REGN3500)	NCT03738423	Phase IIb for AD	Terminated (Lack of efficacy)

**Table 3 biomedicines-10-02700-t003:** Clinical trials targeting Interleukin 1α in AD.

Target Molecule	Clin Trial Gov	Type of Study	Status
**Interleukin 1α**			
Anti-Interleukin 1α antibody Bermekimab (MABp1)	NCT03496974	Phase II for AD	Completed
Anti-Interleukin 1α antibody bermekimab (JNJ-77474462)	NCT04791319	Phase IIb for AD	Terminated (Premature Termination due to Interim Analysis (100 patients at Week 16) meeting futility)

**Table 4 biomedicines-10-02700-t004:** Clinical trials targeting AhR in AD.

Target Molecule	Clin Trial Gov	Type of Study	Status
**AhR**			
AhR agonist, tapinarof 1% topical small molecule	NCT05186805	Phase II for AD	Recruiting
AhR agonist, tapinarof 1% topical small molecule	NCT05142774	Phase III for AD	Recruiting
AhR agonist, tapinarof 1% topical small molecule	NCT05014568	Phase III for AD	Recruiting
AhR agonist, tapinarof 1% topical small molecule	NCT05032859	Phase III for AD	Recruiting

**Table 5 biomedicines-10-02700-t005:** Clinical trials targeting OX40 in AD.

Target Molecule	Clin Trial Gov	Type of Study	Status
**OX40**			
Anti-OX40 antibody KHK4083	NCT03703102	Phase II for AD	Completed
Anti-OX40 antibody ISB 830	NCT03568162	Phase II for AD	Completed
**OX40L**			
Anti-OX40L antibody KY1005	NCT03754309	Phase II for AD	Completed
Anti-OX40L antibody	NCT05131477	Phase II for AD	Recruiting

**Table 6 biomedicines-10-02700-t006:** Clinical trials targeting PDE4 in AD.

Target Molecule	Clin Trial Gov	Type of Study	Status
**PDE4**			
PDE4 inhibitor, Crisaborole 2% topical small molecule	NCT04023084	Phase IV	Completed
PDE4 inhibitor, Crisaborole 2% topical small molecule	NCT04214197	Phase IV	Active, not recruiting
PDE4 inhibitor, Crisaborole 2% topical small molecule	NCT04800185	Early Phase I	Active, not recruiting
PDE4 inhibitor, Crisaborole 2% topical small molecule	NCT03233529	Phase II	Completed
PDE4 inhibitor, Roflumilast 0.15% ARQ-151 Active topical small molecule INTEGUMENT-I	NCT04773587	Phase III	Recruiting
PDE4 inhibitor, Roflumilast 0.15% ARQ-151 Active topical small molecule INTEGUMENT-II	NCT04773600	Phase III	Recruiting
PDE4 inhibitor, Roflumilast 0.05% ARQ-151 Active topical small molecule INTEGUMENT-PED	NCT04845620	Phase III	Recruiting
PDE4 inhibitor, Roflumilast 0.05–0.15% ARQ-151 Active topical small molecule INTEGUMENT-OLE	NCT04804605	Phase III	Recruiting
PDE4 inhibitor, Roflumilast 0.05% topical small molecule	NCT01856764	Phase II	Completed
PDE4 inhibitor, Hemay808 1%/3%/7% topical small molecule	NCT04352595	Phase II	Unknown
PDE4 inhibitor, GW842470X 3% **Topical small molecule**	NCT00354510	Phase II	Completed
PDE4 inhibitor, GW842470X 3% topical small molecule	NCT00356642	Phase I	Completed
PDE4 inhibitor, Apremilast 20 mg **oral** small molecule	NCT00931242	Phase II	Completed
**PDE4 subtype with high PD4 affinity**			
PDE4 inhibitor, Orismilast 20 mg, 30 mg, or 40 mg **oral** small molecule	NCT05469464	Phase II	Recruiting

**Table 7 biomedicines-10-02700-t007:** Clinical trials targeting IL-22 in AD.

Target Molecule	Clin Trial Gov	Type of Study	Status
**IL-22**			
Anti-IL-22 antibody Fezakinumab (ILV-094)	NCT01941537	Phase II	Completed
**IL-22R1**			
Anti-IL-22R1 antibody LEO 138559	NCT04922021	Phase II for AD	Active, not recruiting
Anti-IL-22R1 antibody LEO 138559	NCT03514511	Phase I	Completed
Anti-IL-22R1 antibody LEO 138559	NCT05099133	Phase I	Completed
Anti-IL-22R1 antibody LEO 138559	NCT05470114	Phase II	Recruiting

**Table 8 biomedicines-10-02700-t008:** Clinical trials targeting IL-17c in AD.

Target Molecule	Clin Trial Gov	Type of Study	Status
**IL-17c**			
Anti-Interleukin 17c antibody (MOR 106)	NCT03568071	Phase II for AD	Terminated (MOR106 clinical development in atopic dermatitis was stopped for futility)
Anti-Interleukin 17c antibody (MOR 106)	NCT03689829	Phase I for AD	Terminated (MOR106 clinical development in atopic dermatitis was stopped for futility)

**Table 9 biomedicines-10-02700-t009:** Clinical trials targeting H4R in AD.

Target Molecule	Clin Trial Gov	Type of Study	Status
**Histamine 4 receptor H4R**			
H4R antagonist Oral small molecule 30 mg 50 mg (ZPL389) adriforant	NCT03948334	Phase II for AD	Terminated (Core terminated due to lack of efficacy)
H4R antagonist Oral small molecule 3 mg, 10 mg, 30 mg or 50 mg (ZPL389) adriforant	NCT03517566	Phase II for AD	Terminated (Lack of efficacy)
H4R antagonist oral small molecule (LEO 152020)	NCT05117060	Phase II for AD	Recruiting
H4R antagonist oral small molecule 100 mg, 300 mg (JNJ 39758979)	NCT01497119	Phase II for AD	Terminated (This study was terminated prematurely due to two cases of agranulocytosis.)

**Table 10 biomedicines-10-02700-t010:** Clinical trials targeting IL-31 in AD.

Target Molecule	Clin Trial Gov	Type of Study	Status
**IL-31**			
Anti-IL-31Rα monoclonal antibody Nemolizumab	NCT04921345	Phase II for AD	Recruiting
Anti-IL-31Rα monoclonal antibody Nemolizumab	NCT04562116	Phase II for AD	Recruiting
Anti-IL-31Rα monoclonal antibody Nemolizumab	NCT03921411	Phase II for AD	Completed
Anti-IL-31Rα monoclonal antibody Nemolizumab	NCT03985943	Phase III for AD	Active, not recruiting
Anti-IL-31Rα monoclonal antibody Nemolizumab	NCT03989349	Phase III for AD	Active, not recruiting
Anti-IL-31Rα monoclonal antibody Nemolizumab	NCT03989206	Phase III for AD	Recruiting
Anti-IL-31Rα monoclonal antibody Nemolizumab	NCT03100344	Phase II for AD	Completed
Anti-IL-31Rα monoclonal antibody Nemolizumab	NCT04365387	Phase II for AD	Recruiting
Anti-IL-31Rα monoclonal antibody Nemolizumab	NCT01986933	Phase II for AD	Completed
Anti-IL-31Rα monoclonal antibody Nemolizumab	NCT04204616	Phase III for PN	Recruiting
Anti-IL-31Rα monoclonal antibody Nemolizumab	NCT04501679	Phase III for PN	Completed
Anti-IL-31Rα monoclonal antibody Nemolizumab	NCT05052983	Phase III for PN	Recruiting
Anti-IL-31Rα monoclonal antibody Nemolizumab	NCT04501666	Phase III for PN	Recruiting
Anti-IL-31Rα monoclonal antibody Nemolizumab	NCT03181503	Phase II for PN	Completed
**Interleukin 31 and oncostatin-M (OSM)**			
Anti-OSMRβ monoclonal antibody Vixarelimab (KPL-716)	NCT03816891	Phase II for PN	Recruiting

**Table 11 biomedicines-10-02700-t011:** Clinical trials targeting substance P-NK-1R in AD.

Target Molecule	Clin Trial Gov	Type of Study	Status
**Substance P-NK-1R**			
Neurokinin-1 Receptor Antagonist Oral small molecule Tradipitant (VLY-686)	NCT04140695	Phase III for AD	Completed
Neurokinin-1 Receptor Antagonist Oral small molecule Tradipitant (VLY-686)	NCT03568331	Phase III for AD	Completed
Neurokinin-1 Receptor Antagonist Oral small molecule Serlopitant 1 g or 5 mg (VPD-737)	NCT02975206	Phase II for AD	Completed
Neurokinin-1 Receptor Antagonist Oral small molecule Serlopitant 5 mg (VPD-737)	NCT02196324	Phase II for PN	Completed
Neurokinin-1 Receptor Antagonist Oral small molecule Serlopitant 5 mg (VPD-737)	NCT03540160	Phase III for Pruritus	Terminated (No longer pursuing the development of serlopitant)

**Table 12 biomedicines-10-02700-t012:** Clinical trials targeting P2X3 receptor in AD.

Target Molecule	Clin Trial Gov	Type of Study	Status
**P2X3 receptor**			
P2X3 antagonist Oral small molecule (BLU- 5937),	NCT04693195	Phase II for AD	Completed

**Table 13 biomedicines-10-02700-t013:** Clinical trials targeting IgE in AD.

Target Molecule	Clin Trial Gov	Type of Study	Status
**IgE**			
Anti-IgE monoclonal antibody Omalizumab	NCT01179529	Phase II for AD	Completed
Anti-IgE monoclonal antibody Omalizumab	NCT01678092	Phase I for AD	Completed
Anti-IgE monoclonal antibody Omalizumab	NCT00822783	Phase IV for AD	Completed
Anti-IgE monoclonal antibody Omalizumab	NCT02300701	Phase IV for AD	Completed
Anti-IgE monoclonal antibody Omalizumab	NCT00367016	Phase IV for AD, allergic rhinitis, asthma	Completed
**CεmX domain of membrane-bound IgE**			
Anti-CεmX monoclonal antibody FB825	NCT04413942	Phase II For AD	Active, not recruiting

**Table 14 biomedicines-10-02700-t014:** Clinical trials targeting IL-4 e IL-13 in AD.

Target Molecule	Clin Trial Gov	Type of Study	Status
**IL-4Rα**			
Anti-IL-4α monoclonal antibody Dupilumab		FDA approval for AD March 2017	
Anti-IL-4α monoclonal antibody CBP-201	NCT04444752	Phase II for AD	Completed
Anti-IL-4α monoclonal antibody CBP-201	NCT05017480	Phase II for AD	Recruiting
Anti-IL-4α monoclonal antibody AK120 (Akesobio)	NCT04256174	Phase Ib for AD	Completed
Anti-IL-4α monoclonal antibody AK120 (Akesobio)	NCT05048056	Phase II for AD	Recruiting
**IL-13Rα1**			
Anti-IL-13Rα1 monoclonal antibodyASLAN004 (ASLAN)	NCT04090229	Phase Ib for AD,	Completed
Anti-IL-13Rα1 monoclonal antibodyASLAN004 (ASLAN)	NCT05158023	Phase II for AD	Recruiting
**IL-13**			
Anti-IL-13 monoclonal antibody Tralokinumab		FDA approval for AD December 2021	
Anti-IL-13 monoclonal antibody Lebrikizumab	NCT03443024	Phase II for AD	Completed
Anti-IL-13 monoclonal antibody Lebrikizumab	NCT04178967	Phase III for AD	Completed
Anti-IL-13 monoclonal antibody Lebrikizumab	NCT04146363	Phase III for AD	Completed
Anti-IL-13 monoclonal antibody Lebrikizumab	NCT04250337	Phase III for AD	Completed
Anti-IL-13 monoclonal antibody Lebrikizumab	NCT04392154	Phase III for AD	Recruiting

**Table 15 biomedicines-10-02700-t015:** Clinical trials targeting IL-5 in AD.

Target Molecule	Clin Trial Gov	Type of Study	Status
**IL-5**			
Anti-IL-5 monoclonal antibody mepolizumab	NCT03055195	Phase II for AD	Terminated (This study reached the pre-determined futility criteria following interim analysis. No safety concerns were noted)
**IL-5Rα**			
Benralizumab Monoclonal antibody anti-IL-5Rα	NCT03563066	Phase II for AD	Completed
Benralizumab Monoclonal antibody anti-IL-5Rα	NCT04605094	Phase II for AD	Active, not recruiting

**Table 16 biomedicines-10-02700-t016:** Summary of Janus kinase inhibitors in clinical trials for atopic dermatitis.

JAK Inhibitor (Administration)	Phase ^1^	Approved for AD	JAK1	JAK2	JAK3	TYK2	SYK
Abrocitinib (oral)	3	Yes (FDA and EMA)	Yes	-	-	-	-
Upadacitinib (oral)	3	Yes (FDA and EMA)	Yes	-	-	-	-
Baricitinib (oral)	3	Yes (FDA and EMA)	Yes	Yes	-	-	-
Ruxolitinib (topical)	3	Yes (FDA)	Yes	Yes	-	-	-
Tofacitinib (topical)	2a	No	Yes	Yes	Yes	-	-
Delgocitinib (topical)	3 ^2^	No ^2^	Yes	Yes	Yes	Yes	-
Gusacitinib (oral)	2b	No	Yes	Yes	Yes	Yes	Yes

JAK, Janus kinase; SYK, spleen tyrosine kinase; TYK2, tyrosine kinase 2. ^1^ Furthest phase achieved in a complete clinical trial; ^2^ clinical use in Japan.

**Table 17 biomedicines-10-02700-t017:** Results from complete studies of the newest systemic drugs used for the treatment of atopic dermatitis.

Systemic Molecule	Results
**TSLP**	
Anti-TSLP antibody Tezepelumab (AMG-157/MEDI9929) Phase IIa for AD	A numerically greater percentage of tezepelumab plus TCS-treated patients achieved EASI50 (64.7%) versus the placebo plus TCS (48.2%; P = 0.091). Not statistically significant, numerical improvements over the placebo for all week 12 endpoints were demonstrated, with greater week 16 responses.
**IL-1α**	
Anti-Interleukin 1α antibody Bermekimab (MABp1) Phase II for AD	Acceptable safety profile. Efficacy results for the highest dose indicated that 39% of patients reached the status of clear or almost clear (IGA 0/1). Reduction in itching, with a 68% improvement in the pruritus numerical rating scale (NRS).
**OX40**	
Anti-OX40 antibody KHK4083 Phase II for AD	Efficacy results: 74% reduction in EASI score; IGA 0/1 was reached in 35% of patients.
Anti-OX40 antibody ISB 830 Phase II for AD	Well tolerated. Efficacy results showed that EASI50 was reached in 78% of patients who received the antibody, compared with 38% in the placebo group.
**OX40L**	
Anti-OX40L antibody KY1005 Phase II for AD	Change in EASI from the baseline of 80.1% versus 49.4% in the placebo group. IGA 0/1 was reached in 44% of those who received KY1005 versus 8% in the placebo group.
**IL-22**	
Anti-IL-22 antibody Fezakinumab (ILV-094) Phase II	No significant difference in the change in SCORAD compared with the baseline.
**IL-17c**	
Anti-Interleukin 17c antibody (MOR 106) Phase II For AD	Terminated (MOR106 clinical development in atopic dermatitis was stopped for futility). Adverse events primarily included acneiform lesions in some patients.
**Histamine 4 receptor H4R**	
H4R antagonist Oral small molecule 30 mg 50 mg (ZPL389) adriforant Phase II	Terminated (Core terminated due to a lack of efficacy).
H4R antagonist oral small molecule 100 mg, 300 mg (JNJ 39758979) Phase II	Terminated. (This study was terminated prematurely due to two cases of agranulocytosis.)
**IL-31**	
Anti-IL-31Rα monoclonal antibody Nemolizumab Phase II	Significant decrease in the pruritus sensation measured by a decrease in PP-NRS (69% in the treatment arm versus 34% in the placebo group). IGA 0/1 was reached in 33% of the patients versus 12% in the placebo group at week 16.
**Substance P- NK-1R**	
Neurokinin-1 Receptor Antagonist Oral small molecule Tradipitant (VLY-686)	A reduction of 50% in SCORAD from the baseline was noticed in mild forms of AD.
Neurokinin-1 Receptor Antagonist Oral small molecule Serlopitant 1,g or 5 mg (VPD-737)	Missed the primary endpoint of change in the worst itch NRS from the baseline.
**P2X3 receptor**	
P2X3 antagonist Oral small molecule (BLU-5937),	No results posted.
**IgE**	
Anti-IgE monoclonal antibody Omalizumab Phase IV	Clinical improvement was shown in a small series of patients. The best results in terms of efficacy on eczema have been recorded in patients with total IgE serum levels not exceeding 700 IU/mL.
**IL-5**	
Anti-IL-5 monoclonal antibody Mepolizumab Phase II	Terminated. (This study reached its pre-determined futility criteria following interim analysis. No safety concerns were noted.)
**IL-5Rα**	
Benralizumab Monoclonal antibody anti-IL-5Rα Phase II	No results posted.
**IL-4Rα**	
Anti-IL-4α monoclonal antibody CBP-201	Efficacy results: It seemed to have a faster onset of action. After 4 weeks of therapy, IGA 0/1 was seen in up to 50% of patients receiving CBP-201 versus 13% in the placebo group. The mean reduction in EASI from the baseline was 74% versus 33% in the placebo group.
Anti-IL-4α monoclonal antibody AK120 (Akesobio)	No results posted.
**IL-13Rα1**	
Anti-IL-13Rα1 monoclonal antibody ASLAN004 (ASLAN) Phase Ib	Interim data analysis showed that the compound was well tolerated and provided promising efficacy data, with 67% of the patients achieving EASI75 versus 0% in the placebo group.
**IL-13**	
Anti-IL-13 monoclonal antibody Lebrikizumab Phase IIb	Efficacy results of monotherapy: Dose-dependent improvement in the EASI percentage change from the baseline to week 16 compared to the placebo (125 mg every 4 weeks: −62%; 250 mg every 4 weeks: −69%; 250 mg every 2 weeks: −72%). Lebrikizumab was well tolerated, with a low risk of AEs, including injection-site reactions, herpesvirus infection and conjunctivitis
**JAK inhibitors**	
JAK 1 inhibitor Abrocitinib Oral small molecule Phase III	Efficacy results at W12: More patients in the abrocitinib 100 and 200 mg groups than in the placebo group achieved an IGA of clear or near-clear (24%, 44% and 8%, respectively) and an EASI 75 response (40%, 63% and 12%, respectively). Safety results: Adverse events, including the exacerbation of atopic dermatitis, nasopharyngitis, nausea and headaches, were reported in 69 and 78% of patients in the 100 and 200 mg abrocitinib groups, respectively, and in 57% of patients in the placebo group.
JAK 1 inhibitor Upadacitinib Oral small molecule Phase III trials	Efficacy results at W16 in monotherapy: More patients in the upadacitinib groups than in the placebo groups achieved the primary endpoint of EASI 75 (60% and 70% in the 15 mg groups and 73% and 80% in the 30 mg groups compared with 47% and 46% in the placebo groups, respectively). Efficacy results at W16 in association with moderate-potency topical corticosteroids: More patients in the upadacitinib 15 and 30 mg plus topical corticosteroid groups than in the placebo plus topical corticosteroid group achieved EASI 75 (65%, 77% and 26%, respectively). Safety results: The most frequently reported adverse events were acne upper respiratory tract infections, nasopharyngitis, headaches, creatine phosphokinase elevation and worsening atopic dermatitis.
JAK (1-2) inhibitor Baricitinib Oral small molecule Phase III trials monotherapy (BREEZE-AD1 and BREEZE-AD2)	Efficacy results at W16: More patients treated with baricitinib 2 mg and 4 mg daily as a monotherapy achieved a validated IGA score of 0/1 (clear or near clear) than the placebo (11.4%, 16.8% and 4.8%, respectively, in BREEZE-AD1; 10.6%, 13.8% and 4.5%, respectively, in BREEZE-AD2). Adverse events occurred in about 60% of patients in all groups; the most frequent adverse events reported in the baricitinib groups were nasopharyngitis and headaches.
SYK-JAK inhibitor Oral small molecule Gusacitinib ASN002 Phase II	Efficacy results: Percentage of patients achieving EASI 50 at different dosages vs. placebo: 20 mg 20%, P = 0.93; 40 mg 100%, P = 0.003; 80 mg 83%, P = 0.03; placebo 22%. Percentage of patients achieving EASI 75 at different dosages vs. placebo 20 mg 0%, P = 0.27; 40 mg 71%, P = 0.06; 80 mg 33%, P = 0.65; placebo 22%. Change from the baseline in itching at different dosages vs. placebo: 20 mg −1–3 ± 2–1, P = 0.81; 40 mg −3–1 ± 2–7, P = 0.27; 80 mg −4–7 ± 2–1, P = 0.01; placebo −1–6 ± 1–8. Safety results: Gusacitinib was well tolerated at all dosages. Adverse events were generally mild to moderate and of short duration. The most common forms of treatment were for upper respiratory tract infections, headaches, nausea and nasopharyngitis. Two AE meet the stopping rule, i.e., mild hypertension and low lymphocyte counts. The event of mild hypertension was reported in a patient receiving 80 mg and was classified as a possibly related TEAE. The event of lymphopenia was reported in a patient who had predose lymphocyte levels of 0·67 × 103 μL^−1^ at day 1 and 0·56 × 103 μL^−1^ at day 8. No thromboembolic events or opportunistic infections were reported.

**Table 18 biomedicines-10-02700-t018:** Results from complete studies of the newest topical drugs used for the treatment of atopic dermatitis.

Topical Molecule	Results
**Cutaneous microbiota**	
Three strains of Roseomonas mucosa FB-401 Phase II for AD	Of the adult patients, 60% showed a 50% reduction in the Scoring of Atopic Dermatitis (SCORAD), 90% of the paediatric patients achieved Eczema Area Severity Index (EASI) 50 and 30% achieved EASI 90. Failure of the Phase II study (protocol FB401-01) to meet its endpoint.
**AhR**	
AhR agonist, tapinarof 1% topical small molecule Phase II for AD	Efficacy results showed that 53% of the patients reached the primary endpoint of Investigator Global Assessment (IGA) 0/1 versus 28% in the placebo group.
**PDE4**	
PDE4 inhibitor, Crisaborole 2% topical small molecule Phase III AD-301:NCT02118766; AD-302: NCT02118792;	Efficacy results: IGA score of clear or almost clear (AD-301: 51.7% vs. 40.6%, *P* = 0.005; AD-302: 48.5% vs. 29.7%, *P* < 0.001). Sustained relief from pruritus. Crisaborole 2% ointment resulted in a clinically meaningful improvement in the QoL of patients and their families.
PDE4 inhibitor, Roflumilast 0.05% topical small molecule Phase II	Failed to reach the primary endpoint (change in EASI from the baseline). More Phase III studies are ongoing, but the results are not available.
**JAK inhibitors**	
JAK1 and 2 inhibitor Ruxolitinib Phase III	Efficacy results at W8: A higher percentage of patients treated with ruxolitinib 0.75% cream or ruxolitinib 1.5% cream achieved the primary endpoint of an IGA score of 1 or 2 (clear or near clear) in both studies compared to the placebo (50% and 39%, respectively, in the 0.75 percent cream groups; 54% and 51%, respectively, in the 1.5 percent cream groups; and 15 and 8 percent, respectively, in the vehicle group). More patients in both ruxolitinib groups than in the vehicle group achieved a 75% reduction in Eczema Area and Severity Index (EASI 75) and a clinically relevant reduction in pruritus. Safety results: Burning and itching at the application site, upper respiratory tract infection and headaches were the most common adverse events. No adverse events were suggestive of systemic JAK1/JAK2 kinase inhibition.
JAK1,2 and 3 inhibitor Tofacitinib Phase IIa	Efficacy results at W4: The mean percent change from the baseline in the EASI score was significantly greater in patients treated with topical tofacitinib 2% than in those treated with placebo (−82% and −30%, respectively). Safety results: Adverse effects, including infection, increased blood creatine phosphokinase and contact dermatitis, were mild and occurred in 31% of patients treated with tofacitinib and 60% of those treated with the placebo.

## Data Availability

Not applicable.

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
