# Peer review of "A Systematic Review of Atopic Dermatitis: The Intriguing Journey Starting from Physiopathology to Treatment, from Laboratory Bench to Bedside"

_biomedicines, 2022, doi:10.3390/biomedicines10112700_

Round 1
Reviewer 1 Report
This manuscript describes the increasing knowledge on atopic dermatitis focusing on emerging therapeutic targets.
The methodology adopted is adequate, the results interesting, and the reading of the manuscript is understandable and exhaustive, some minor suggestions:
-Although the topic focuses on innovative therapeutic targets, I suggest the authors to add a short paragraph on IL4-IL targets13, although it is a known target, it currently represents a milestone in the treatment of patients.
-Before using an acronym authors should write the full name of the molecule: antimicrobial peptides (AMPs ); interleukin (IL)-4; (dendritic cells (DCs); Toll-like receptor 5 (TLR5) and tumor necrosis factor receptor (TNFR), furthermore, after having specified the extension of the acronym, it is recommended to use it exclusively.
- Please, place the first table after paragraph 5.1
Author Response
Reviewer 1
This manuscript describes the increasing knowledge on atopic dermatitis focusing on emerging therapeutic targets.
The methodology adopted is adequate, the results interesting, and the reading of the manuscript is understandable and exhaustive, some minor suggestions:
Q:-Although the topic focuses on innovative therapeutic targets, I suggest the authors to add a short paragraph on IL4-IL targets13, although it is a known target, it currently represents a milestone in the treatment of patients.
R- a short paragraph focusing on IL4 has been added to the manuscript
Q-Before using an acronym authors should write the full name of the molecule: antimicrobial peptides (AMPs ); interleukin (IL)-4; (dendritic cells (DCs); Toll-like receptor 5 (TLR5) and tumor necrosis factor receptor (TNFR), furthermore, after having specified the extension of the acronym, it is recommended to use it exclusively.
R: full name of all molecules reported into the manuscript has been extensively reported before to adopt exclusively the acronym
Q- Please, place the first table after paragraph 5.1
R: table 1 has been moved after paragraph 5.1
Reviewer 2 Report
This paper is well written, however there are some typo errors.
Scientific names of bacterial species should be written in italics.
It would be better if the authors can give brief results of completed studies in the text or table in order to see which target treatments have higher efficacy.
Please add the reference of a phase II randomized controlled study evaluated the efficacy of the anti-TSLP monoclonal antibody in line 215 and the reference of FceRI on epidermal Langerhans cells as a biomarker for AD in line 414.
Author Response
Reviwer 2
This paper is well written, however there are some typo errors.
Manuscript has been extensively revised in order to remove all typos.
Scientific names of bacterial species should be written in italics.
Scientific names of bacterial species should be reported in italics.
It would be better if the authors can give brief results of completed studies in the text or table in order to see which target treatments have higher efficacy.
A table including briefly the results of completed studies has been added to the manuscript.
Please add the reference of a phase II randomized controlled study evaluated the efficacy of the anti-TSLP monoclonal antibody in line 215 and the reference of FceRI on epidermal Langerhans cells as a biomarker for AD in line 414.
Specific references have been reported.